# HER2 Status in High-Risk Endometrial Cancers (PORTEC-3): Relationship with Histotype, Molecular Classification, and Clinical Outcomes

**DOI:** 10.3390/cancers13010044

**Published:** 2020-12-25

**Authors:** Lisa Vermij, Nanda Horeweg, Alicia Leon-Castillo, Tessa A. Rutten, Linda R. Mileshkin, Helen J. Mackay, Alexandra Leary, Melanie E. Powell, Naveena Singh, Emma J. Crosbie, Vincent T.H.B.M. Smit, Carien L. Creutzberg, Tjalling Bosse

**Affiliations:** 1Department of Pathology, Leiden University Medical Center, 2300 RC Leiden, The Netherlands; L.Vermij@lumc.nl (L.V.); A.Leon_del_Castillo@lumc.nl (A.L.-C.); T.A.Rutten@lumc.nl (T.A.R.); V.T.H.B.M.Smit@lumc.nl (V.T.H.B.M.S.); 2Department of Radiation Oncology, Leiden University Medical Center, 2300 RC Leiden, The Netherlands; N.Horeweg@lumc.nl (N.H.); C.L.Creutzberg@lumc.nl (C.L.C.); 3Division of Cancer Medicine, Peter MacCallum Cancer Centre, Melbourne, VIC 3000, Australia; Linda.Mileshkin@petermac.org; 4Division of Medical Oncology and Hematology, Sunnybrook Odette Cancer Centre, Toronto, ON M4N 3M5, Canada; helen.mackay@sunnybrook.ca; 5Department of Medical Oncology, Gustave Roussy, 94805 Villejuif, France; Alexandra.LEARY@gustaveroussy.fr; 6Department of Clinical Oncology, Barts Health NHS Trust, London E1 1BB, UK; melanie.powell10@nhs.net; 7Department of Pathology, Barts Health NHS Trust, London E1 1BB, UK; N.Singh@bartsandthelondon.nhs.uk; 8Division of Cancer Sciences, University of Manchester, St Mary’s Hospital, Manchester M13 9WL, UK; Emma.Crosbie@manchester.ac.uk; 9Department of Obstetrics and Gynecology, Manchester University NHS Foundation Trust, Manchester Academic Health Science Centre, Manchester M13 9NQ, UK

**Keywords:** endometrial cancer, high-risk, HER2, *ERBB2*, p53

## Abstract

**Simple Summary:**

HER2 testing in endometrial cancer (EC) has gained renewed interest as a therapeutic target. However, HER2 status has not been investigated in the context of the molecular EC classification. Here, we aimed to determine the clinicopathological features and prognostic significance of the HER2 status in the molecularly classified PORTEC-3 trial population of patients with high-risk EC. HER2 status of 407 high-risk EC was determined by HER2 immunohistochemistry and HER2 dual in situ hybridization. Twenty-four (5.9%) HER2-positive EC of various histological subtypes were identified, including serous (*n* = 9, 37.5%), endometrioid (*n* = 6, 25.0%), and clear cell (*n* = 5, 20.8%). HER2 positivity was highly associated with the p53-abnormal subgroup (p53abn, 23/24 cases; *p* < 0.0001). The correlation between p53abn and the HER2 status (ρ = 0.438; *p* < 0.0001) was significantly stronger (*p* < 0.0001) than between serous histology and the HER2 status (ρ = 0.154; *p* = 0.002). HER2 status did not have independent prognostic value for survival after correction for the molecular classification. Our study strongly suggests that molecular subclass-directed HER2 testing is superior to histotype-directed testing.

**Abstract:**

HER2 status has not been investigated in the context of the molecular endometrial cancer (EC) classification. Here, we aimed to determine the clinicopathological features and prognostic significance of the HER2 status in the molecularly classified PORTEC-3 trial population of patients with high-risk EC (HREC). HER2 testing was performed on tumor tissues of 407 molecularly classified HREC. HER2 status was determined by HER2 immunohistochemistry (IHC; all cases) and subsequent HER2 dual in situ hybridization for cases with any (in) complete moderate to strong membranous HER2 IHC expression. The Χ^2^ test and Spearman’s Rho correlation coefficient were used to compare clinicopathological and molecular features. The Kaplan–Meier method, log-rank test, and Cox proportional hazards models were used for survival analysis. We identified 24 (5.9%) HER2-positive EC of various histological subtypes including serous (*n* = 9, 37.5%), endometrioid (*n* = 6, 25.0%), and clear cell (*n* = 5, 20.8%). HER2 positivity was highly associated with the p53-abnormal subgroup (p53abn, 23/24 cases; *p* < 0.0001). The correlation between p53abn and the HER2 status (ρ = 0.438; *p* < 0.0001) was significantly stronger (*p* < 0.0001) than between serous histology and the HER2 status (ρ = 0.154; *p* = 0.002). HER2 status did not have independent prognostic value for survival after correction for the molecular classification. Our study strongly suggests that molecular subclass-directed HER2 testing is superior to histotype-directed testing. This insight will be relevant for future trials targeting HER2.

## 1. Introduction

Endometrial cancer (EC) is the most common gynecological cancer in developed countries and is primarily treated with surgery. Around 15–20% of women with EC have high-risk disease with increased incidence of distant metastases and cancer-related death. High-risk features include advanced age, high-grade non-endometrioid histology, substantial lymphovascular space invasion (LVSI), and advanced stage disease. For many years, standard adjuvant treatment for women with high-risk EC has been pelvic external beam radiotherapy (EBRT) [1]. The international randomized Adjuvant Chemoradiotherapy Versus Radiotherapy Alone in Women with High-Risk Endometrial Cancer (PORTEC-3) trial investigated the benefit of adjuvant chemotherapy in combination with EBRT (chemoradiotherapy (CTRT)) versus EBRT alone (radiotherapy (RT)) in patients with high-risk EC [2]. A small but significant benefit was shown in overall survival (OS) and failure-free survival (FFS) in favor of the CTRT treatment arm, albeit at the cost of significantly toxicity. Thus, the identification of patients who will benefit from chemotherapy is essential. However, the assessment of high-risk pathologic features is subject to a substantial interobserver disagreement, complicating the identification of this subset of patients [3,4].

Over the last decade, the Cancer Genome Atlas (TCGA)-based molecular EC classification has proven to have significant prognostic value in EC patients with the potential to refine current clinicopathological risk assessment [5,6,7,8]. The molecular classification stratifies EC into four distinct molecular subgroups including: (1) *POLE*-ultramutated (*POLE*mut) EC which are characterized by pathogenic mutations in the exonuclease domain of *POLE*, associated with endometrioid histology, and have an excellent prognosis, (2) mismatch repair-deficient (MMRd) EC which are characterized by microsatellite instability, are exclusive to endometrioid histology, and show an intermediate prognosis, (3) p53-abnormal (p53abn) EC which are associated with somatic copy number alterations (SCNA), high-grade endometrioid and serous histology, and have a poor prognosis, (4) no-specific-molecular-profile (NSMP) EC which are microsatellite-stable and have low SCNA, associated with low-grade endometrioid histology and have an intermediate prognosis. Furthermore, the molecular classification provides a strong biological basis for future subclass-specific clinical trials. Defining therapeutic targets within the p53abn EC subgroup should be prioritized given its poor prognosis despite adjuvant CTRT [9].

Human epidermal growth factor receptor 2 (HER2), encoded by the *ERBB2* gene, has been established as an important biomarker with both prognostic and therapeutic implications in breast and gastric cancers [10,11,12]. HER2-positive breast cancers are associated with aggressive disease course, poor prognosis, and show a strong correlation with other independent prognostic markers such as histologic subtype and grade [13]. Trastuzumab, a monoclonal antibody directed against HER2, has proven to increase survival of patients with HER2-positive breast cancer and of those with advanced and metastatic gastric cancer. This has subsequently led to the approval of trastuzumab therapy in these cancers by the Food and Drug Association (FDA) [14,15,16,17].

HER2 protein overexpression and/or *ERBB2* amplifications have also been described in EC, albeit with conflicting estimates of frequencies and clinical outcomes. Possibly, this is due to a lack of universal HER2 testing and scoring methods and differences in histological subtypes included in the studies [18,19,20,21]. Nevertheless, existing evidence points towards a correlation between HER2 protein overexpression and/or *ERBB2* amplification in EC with serous histology (SEC), with reported frequencies ranging between 29–49% [18,19,20,21]. Recently, a phase II clinical trial comparing adjuvant carboplatin–paclitaxel with and without trastuzumab in 58 patients with advanced and recurrent HER2-overexpressing SEC showed increased progression-free survival in favor of the combined chemotherapy–trastuzumab treatment arm [22,23]. These findings encourage further investigation of the efficacy of trastuzumab in combination with chemotherapeutic agents in HER2-overexpressing/amplified EC.

Although the clinical and therapeutic implications of HER2 overexpression/amplification in EC have been studied over the last decade, it has not yet been investigated in the context of the molecular EC classification. We hypothesize that molecular subgroup-directed screening for HER2 status is more effective in identifying patients who may benefit from anti-HER2 therapies than histology-directed screening for HER2 status. In addition, the presence of *PIK3CA* mutations and/or *PTEN* loss have been associated with trastuzumab resistance in HER2-positive breast cancers, [24,25,26]. Previous studies investigating the efficacy of trastuzumab in HER2-overexpressing EC did not take this into account, possibly (negatively) influencing clinical trial outcomes. Gaining more insight in the prevalence of *PIK3CA* and *PTEN* mutations in HER2-positive EC can provide essential information about the magnitude of this trastuzumab resistance mechanism in EC.

The molecular EC classification is currently being implemented in clinical practice [27]. Now that anti-HER2 treatments in EC patients show promising results, it is time to study HER2 more comprehensively. Therefore, we aimed to investigate the association between the HER2 amplification status and clinicopathological and molecular features as well as patient outcomes in the well-defined PORTEC-3 trial cohort of high-risk EC supplemented by publicly available TCGA data. To our knowledge, this is the first comprehensive study investigating the correlation between the HER2 amplification status and the molecular EC classification.

## 2. Results

### 2.1. Clinicopathological Characteristics

HER2 status could be determined for 405 (98.8%) molecularly classified ECs from PORTEC-3 (Figure 1). For three cases, there was insufficient material available for HER2 testing. In two cases, HER2 IHC was successful, but the final HER2 status could not be determined due to failed HER2 dual in situ hybridization (DISH) testing and inadequate quality of the copy number plot generated by next-generation sequencing (NGS). The median follow-up at the time of analysis was 6.1 years (95% CI: 5.86–6.27%).

Of all the 405 evaluable cases, 263 EC (64.6%) did not have any membranous HER2 immunoreactivity and 45 EC (11.1%) showed faint membranous HER2 IHC expression. Moderate and strong membranous immunoreactivity was observed in 60 EC (14.7%) and 26 EC (6.4%), respectively. In 13 cases (3.2%), HER2 IHC could not be reliably interpreted due to the absence of a positive external control. After performing DISH on the latter three groups, we identified 24 HER2-positive EC (5.9%; 95% CI: 3.6–8.2%) (Figure 1). Subclonal HER2 expression was observed in 81 cases (20%), including 11 HER2-positive EC (45.8%). Incomplete membranous immunoreactivity was seen in 101 tumors (24.9%), including 6 HER2-positive EC (25.0%). All HER2 IHC slides were additionally scored by an adjusted version of the American Society of Clinical Oncology and College of American Pathologists (ASCO/CAP) 2007 guideline for breast cancer. The HER2 IHC results were compared to the HER2 amplification status by DISH (data shown in Appendix A).

Age, histology, and tumor grade differed significantly between patients with HER2-positive and HER2-negative EC (Table 1). Interestingly, the majority of the HER2-positive EC were of non-serous histology, including endometrioid EC (*n* = 6; 25.0%) and clear cell EC (*n* = 5; 20.8%) (Figure 2). There were no significant differences between HER2-positive and HER2-negative cases in stage, presence of LVSI, and adjuvant treatment received (RT vs. CTRT).

### 2.2. Association with Molecular Classification and Other Molecular Alterations

The prevalence of HER2-positive EC both in the PORTEC-3 as well as in the TCGA cohort was not evenly distributed between the four molecular subgroups (*p* < 0.0001) (Table 2). In PORTEC-3, all but one (95.8%) of the HER2-positive ECs were classified as p53abn. Given this strong association, we additionally performed DISH on all p53abn EC with absent or any faint membranous staining to ascertain the HER2 status assignment by IHC was correct. In correspondence to the IHC results, none of these p53abn EC showed evidence of HER2 amplification. The only HER2-positive case in PORTEC-3 which was not p53abn was an NSMP EC. Examination of this case revealed a convincing wildtype p53-IHC pattern and no *TP53* mutation was found by NGS analysis. The tumor showed moderate incomplete membranous HER2 immunoreactivity in >10% of the tumor and low-level HER2 amplification by DISH (HER2:CEP17 ratio: 2.2, average HER2 count: 3.7). All *POLE*mut EC (*n* = 52) and MMRd EC (*n* = 135) were HER2-negative. Within the TCGA cohort, all *ERBB2*-amplified EC were copy number-high (CN-high). NGS analysis was successful in 18 out of the 24 HER2-positive cases in PORTEC-3. In these cases, pathogenic *PIK3CA* mutations were present in five (27.8%) cases and only one HER2-positive tumor (5.6%) showed a pathogenic *PTEN* frameshift mutation (c.309del; p.C105fs). In the TCGA cohort, ten out of the 25 HER2-positive cases (40.0%) had a pathogenic *PIK3CA* mutation, while none of the cases had a *PTEN* mutation. We found a moderately strong positive correlation between HER2-positivity and the p53abn subgroup (ρ = 0.441, *p* < 0.0001), which explains 19.5% of the shared variance. This correlation was significantly stronger (*p* < 0.0001) than the weak positive correlation that was found between HER2 positivity and serous histology (ρ = 0.154, *p* = 0.002) that only explains 2.4% of the shared variance.

### 2.3. Association between HER2 Status and Clinical Outcome

We analyzed differences in recurrence-free survival (RFS) and OS for women with and without HER2-positive EC in the PORTEC-3 cohort. Patients with HER2-positive EC had significantly poorer survival than patients with HER2-negative EC (5-year RFS: 42.9% vs. 73.6%, *p* < 0.0001; 5-year OS: 42.3% vs. 81.4%, *p* < 0.0001) (Figure 3A,B). However, in a sub-analysis including only p53abn EC, there was no difference in prognosis (Figure 3C,D). The prognostic value of the HER2 status was evaluated in univariable and multivariable analysis including all PORTEC-3 patients (Table 3). In univariable analysis, HER2 positivity was associated with significantly worse recurrence-free survival (HR: 2.76; 95% CI: 1.55–4.93) and overall survival (HR: 3.64; 95% CI: 2.02–6.58). However, when corrected for age, stage, grade, LVSI, and the molecular classification by multivariable analysis, HER2 status was not an independent prognostic factor for both RFS (HR: 1.15; 95% CI: 0.60–2.12) and OS (HR: 1.24; 95% CI: 0.63–2.42).

These analyses were also performed for the combined PORTEC-3 and TCGA cohort confirming the results reported above (Appendix A, Appendix A).

## 3. Discussion

In this study, we demonstrate that HER2 positivity is prevalent in 5.9% (95% CI: 3.6–8.2%) of HREC and arises in various histological subtypes including endometrioid (25.0%), serous (37.5%), and clear cell (20.8%) EC. To our knowledge, this is the first study to investigate HER2 status in the context of the molecular EC classification.

HER2 positivity is strongly associated with the p53abn (*p* < 0.0001) molecular subgroup; 95.8% of all the HER2-positive ECs were identified in this molecular subgroup. This strong association was confirmed in the TCGA cohort; all *ERBB2*-amplified ECs were CN-high (*p* < 0.0001). The association between HER2 positivity and the p53abn subgroup is also reflected in the survival analyses. Significantly lower 5-year RFS and OS were found in patients with HER2-positive versus HER2-negative EC (42.9% vs. 73.6%, *p* < 0.0001 for RFS; 42.3% vs. 81.4%, *p* < 0.0001 for OS). These differences lost their statistical significance when corrected for p53 status (40.2% vs. 51.3%, *p* = 0.313 for RFS; 39.3% vs. 57.4%, *p* = 0.231 for OS). In addition, HER2 status did not have independent prognostic value in multivariable analysis with the established prognostic factors, including the molecular EC classification.

We are the first to report on a highly significant association between HER2 overexpression and/or amplification and the p53abn molecular subgroup (*p* < 0.0001) with 95.8% of HER2-positive EC classified in the p53abn subgroup in the PORTEC-3 HREC. This was confirmed by the similarly strong association between *ERBB2* amplification and the CN-high molecular subgroup (*p* < 0.0001) in the TCGA cohort, with 100% of the *ERBB2*-amplified EC classified in the CN-high subgroup. Importantly, we found that the correlation between HER2 positivity and p53abn EC is significantly stronger than the correlation between HER2 positivity and serous histology (*p* < 0.0001). The interaction between HER2 overexpression and mutant p53 status has been described in other cancers. A study using cancer cell lines with introduced gain-of-function *TP53* mutations found an increase in HER2 mRNA expression levels as well as HER2 protein levels [28]. Analysis of the publicly available breast cancer dataset (GSE22358) confirm significantly higher HER2 mRNA levels in p53 mutant samples (*p* = 0.046) compared to p53 wildtype samples [29]. Mechanistically, mutant p53 has shown to be associated with enhanced transcriptional activity of HSF1 (heat shock transcription factor 1) that targets chaperon Hsp90 which in turn stabilizes the HER2 and p53 proteins, thereby further reinforcing oncogenic signaling [30]. Another study has shown an increased risk of recurrence in breast cancer patients harboring both a somatic *TP53* mutation and *ERBB2* amplification compared to patients with either one or no alterations [31]. Our study is one of the largest studies investigating HER2 amplification status in a cohort of HREC which were not selected on the serous histological subtype. In 2006, Morrison et al. investigated HER2 protein overexpression by IHC and *ERBB2* amplification by FISH in a cohort of 483 ECs [20]. HER2 protein overexpression was reported in 14.2% (69/483) and *ERBB2* amplification was found in 6.6% (32/483) of cases, respectively. Comparable to our findings, 46.9% of all HER2-positive EC tested by FISH were of non-serous histology (9/32: endometrioid, 17/32: serous, 2/32: clear cell carcinomas, and 3/32: other histotypes). In our study, HER2 overexpression and/or amplification was found in 14.1% of serous EC. Other studies including only serous EC reported HER2 overexpression by IHC ranging between 14% and 80% and *ERBB2* amplification by FISH ranging from 21% to 47% [32].

Previously reported frequencies vary widely, possibly due to differences in the composition of study cohorts and differences in the HER2 testing methods used in the absence of an EC-specific HER2 testing method. In our cohort, HER2 amplification status was confirmed by performing DISH on all tumors with any moderate/strong HER2 IHC expression rather than applying a HER2 testing algorithm validated on breast of gastric cancers. Incomplete membranous HER2 expression was observed in 25% (*n* = 6) of HER2-amplified EC. According to the latest ASCO/CAP 2018 HER2 Testing in Breast Cancer Guideline, these cases would have been interpreted as HER2-negative (IHC 1+) without additional in situ hybridization (ISH) testing [33]. This suggests that the current ASCO/CAP guideline for breast cancer may not be directly applicable for the identification of HER2-positive EC. Furthermore, in contrast to breast and gastric HER2 testing guidelines, we did not apply a threshold for the percentage of tumor cells (e.g., >10%, >30%) with moderate/strong HER2 IHC expression required to perform DISH, as such threshold is still arbitrary in EC. It is conceivable that small HER2-positive subclones may still have relevant impact on the biological behavior of a tumor. We foresee that a clinically meaningful threshold for HER2 status determination in EC will evolve from trials using anti-HER2 therapies. Given the small number of cases with <10% HER2 positivity in our study (*n* = 2/24, 8.3%), a threshold of 10% may be a good starting point going forward.

In our study, we did not detect a statistically significant difference in survival between patients with HER2-positive and HER2-negative EC after adjustment for p53 status. However, our study was not powered to detect small differences in prognosis. We can therefore not rule out that HER2 status has minor negative prognostic impact in HREC. Importantly, after extending the PORTEC-3 dataset with the TCGA cohort, we still did not find a significant independent prognostic value of the HER2 status in multivariable analysis (including the molecular classification). This makes it less likely that there is a small prognostic difference that we did not detect. Although previous studies showed prognostic value of HER2 overexpression and/or amplification in EC [19], these findings may have been confounded by an underlying mutant p53 status.

Earlier clinical studies incorporating trastuzumab as a single agent for the treatment of HER2-overexpressing and/or *ERRB2*-amplified EC did not show any prognostic benefit [34,35]. However, a phase II randomized clinical trial comparing carboplatin–paclitaxel with and without trastuzumab in 58 patients with primary stage III/IV or recurrent serous EC showed a significantly prolonged PFS with four months in patients who received combined chemotherapy–trastuzumab treatment [22,23]. These findings encourage further exploration of combined chemotherapy with anti-HER2 therapeutics in HER2-positive EC. Of interest, recently published survival analysis of p53abn EC (of all histotypes) in PORTEC-3 showed a significant benefit in 5-year RFS when chemotherapy was added to adjuvant radiotherapy (5-year RFS of 58.6% with CTRT vs. 36.2% with RT; HR: 0.52, 95% CI: 0.30–0.91), while this was not found for other molecular subgroups [9]. It is possible that the underlying aberrant p53 status in HER2-positive EC (partially) explains the promising results of combined adjuvant chemotherapy–trastuzumab in contrast to single-agent trastuzumab treatment.

Several mechanisms of trastuzumab resistance have been identified in breast cancer, one of which is increased signaling through the PI3-kinase pathway caused by gain-of-function *PIK3CA* mutations or inactivation of *PTEN* [36]. Trastuzumab resistance has been observed in primary HER2-overexpressing serous EC cell lines harboring oncogenic *PIK3CA* mutations [37]. In our study, pathogenic *PIK3CA* mutations were found in 33.9% of the cases (95% CI: 19.8–48.0%; 27.8% and 40.0% of HER2-positive EC in the PORTEC-3 trial and TCGA cohort, respectively). This is comparable to what is observed in breast cancer, in which activating *PIK3CA* mutations are reported in 25–38% of HER2-positive breast cancers [25,38,39]. In contrast, only one HER2-positive case from PORTEC-3 and none of the *ERBB2*-amplified EC had a mutation in *PTEN* suggesting inactivation of *PTEN* will unlikely impact anti-HER2 therapies in HER2-positive EC. Although not much is known about the role of oncogenic *PIK3CA* mutations in trastuzumab resistance in HER2-positive EC, the relatively high prevalence of the co-occurrence may be taken into consideration when designing new clinical trials on trastuzumab.

## 4. Materials and Methods

### 4.1. Patient and Tissue Selection

HER2 testing was performed on formalin-fixed paraffin-embedded (FFPE) tumor tissues molecularly profiled from 410 consenting patients included in the international PORTEC-3 clinical trial and collected by the TransPORTEC group. Detailed information on the study design and results have been described previously [2]. In brief, 660 eligible patients with high-risk EC were randomly assigned 1:1 to postoperative chemoradiotherapy (CTRT) versus radiotherapy (RT) alone. Pathological inclusion criteria for the PORTEC-3 trial were as follows: International Federation of Gynecology and Obstetrics (FIGO) 2009 stage IA grade 3 endometrioid EC (EEC) with documented LVSI; stage IB grade 3 EEC; stage II–III EEC; or non-endometrioid EC with stage IA (with invasion), IB, II, or III. Upfront central pathology review was performed by reference gynecopathologists to confirm eligibility. The study was approved by the Dutch Cancer and medical ethics committees at participating. Patient, tumor, and treatment characteristics were not significantly different between PORTEC-3 trial patients included in and excluded from the molecular analyses, as reported previously [9].

In addition, we used publicly available endometrial cancer data from TCGA series to further extend our cohort [40]. Clinicopathological and molecular data as well as patient outcome data from all patients with available copy number variation (CNV) data were obtained from the Genome Data Analysis Center [41]. 

### 4.2. Immunohistochemical Staining of HER2

Immunohistochemical (IHC) staining of HER2 was performed using Ventana BenchMark GX (Roche Diagnostics, Basel, Switzerland). Available FFPE blocks were cut into 4 μm slides and stained for HER2 IHC using the anti-HER2/neu (4B5) Rabbit Monoclonal Primary Antibody and the Ventana ultraView DAB Detection Kit. Because of differences in fixation time of the tumor specimens, due to the multi-centered character of the PORTEC-3 trial, the cell conditioning step was prolonged, as compared to the standard manufacturer’s protocol, to ensure optimal antibody retrieval for all samples. In addition, the incubation time of hematoxylin counterstaining was prolonged to 8 min for better visualization of cell nuclei. A positive control was mounted on each individual slide.

### 4.3. Evaluation of HER2 IHC Staining

HER2 IHC stained slides were reviewed by an expert gynecopathologist (T.B.)—blinded to clinicopathological and molecular data—to determine the presence and intensity of membranous HER2 immunoreactivity (absent, faint, moderate, and strong) as well as the percentage of tumor cells showing membranous HER2 staining. Representative examples of faint, moderate, and strong membranous staining are shown in Appendix A. All slides with absent HER2 staining—in the presence of a positive control—as well as slides with faint membranous HER2 staining—regardless of the percentage of tumor cells and completeness of membranous staining—were classified as HER2-negative. Based on preliminary results of HER2 IHC and DISH in EC prior to this study as well as experience from diagnostic cases, we felt confident to classify these cases as HER2-negative. To confirm this approach, DISH was performed on a random selection of 20 tumors with absent or faint membranous staining. Tumors with any moderate or strong (in)complete membranous HER2 staining were considered potentially HER2-amplified and final HER2 amplification status of these cases was determined by dual in situ hybridization (DISH).

### 4.4. HER2 Dual In Situ Hybridization

DISH was performed to determine the *HER2* amplification status of all cases with moderate to strong membranous HER2 IHC expression regardless of completeness of membranous staining and the percentage of tumor cells. It has been shown that HER2-amplified EC frequently show incomplete membranous HER2 IHC staining [18]. Therefore, the completeness of membranous staining was not considered a criterium for subsequent DISH testing. In addition, no threshold was applied to the percentage of stained tumor cells with moderate or strong membranous staining needed to perform DISH, as such a threshold is not yet validated in EC. HER2 amplification status of all cases that failed for IHC (*n* = 13) were tested by DISH, as well as a randomly selected subset of tumors with absent (*n* = 10) or faint (*n* = 10) membranous IHC expression to confirm the lack of amplification in cases with these staining patterns. Although validated HER2 testing guidelines exist for directing conformational ISH testing in breast and gastroesophageal cancer, these guidelines have not been properly validated in EC [33,42]. Therefore, for the purpose of this study, we apprehended a low threshold for additional ISH testing with the intention not to miss any HER2-amplified EC. The INFORM HER2 Dual ISH (DISH) DNA Probe Cocktail assay was performed on Ventana BenchMark GX (Roche Diagnostics, Basel, Switzerland). The DISH assay was initially performed using the manufacturer’s recommended protocol. However, a pre-treatment baking step was added to the protocol to prevent nuclear bubbling caused by excessive paraffin. For each slide, visible HER2 probe (black) and CEP17 (red) signals were counted in at least 20 nuclei by standard light microscopy and the HER2:CEP17 ratio was calculated. In line with the ASCO/CAP guideline recommendations for breast cancer, *ERBB2* amplification was defined as the HER2:CEP17 ratio ≥ 2.0 [33]. Since significant heterogeneity of HER2 IHC expression in EC has been reported [43], for every case, the HER2 IHC slide was consulted to identify the area(s) in the tumor in which the HER2:CEP17 ratio needed to be scored in order to prevent possible false-negative results.

### 4.5. Next-Generation Sequencing

For the purpose of previous publications, targeted next-generation sequencing (NGS) using AmpliSeq Cancer Hotspot Panel v5 was performed on PORTEC-3 tumor samples. Detailed description of DNA isolation and sequencing was described previously [9]. In cases with a failed DISH assay, the final HER2 status was determined by visual examination of copy number plots generated for every individual sample for the presence of unequivocal *ERBB2* amplification. All cases with low-quality NGS data were considered ineligible for the examination of *ERBB2* amplification.

The presence of pathogenic *PIK3CA* and *PTEN* mutations was evaluated independent of and blinded for HER2 status. A minimum coverage threshold of 100 reads and variant allele frequency of 0.1 were considered. Pathogenicity of non-synonymous *PIK3CA* and *PTEN* mutations was assessed using the public databases COSMIC [44] and ClinVar [45], as well as in silico tools SIFT [46] and PolyPhen [47]. Only mutations classified as (likely) pathogenic were included in this study.

### 4.6. Statistical Analysis

Statistical analyses were performed with SPSS (Statistical Package of Social Science) version 25 (IBM, Armonk, NY, USA). Associations between groups were analyzed using the χ^2^ statistics or Fisher’s exact test for categorical variables, and the Mann–Whitney U test for continuous variables. The Spearman’s Rho correlation coefficient was calculated to compare the correlations between HER2 amplification and histological and molecular characteristics. The difference between the correlations was tested by transforming the correlations to z-scores; then, we calculated the ratio of the difference to the standard error and compared this ratio to the standard normal distribution [48]. Five-year recurrence-free survival (RFS) and overall survival (OS) were estimated using the Kaplan–Meier’s methodology and compared between groups using the log-rank test. To determine whether HER2 status has prognostic value independent of the established prognostic features, multivariable regression analysis using the Cox proportional hazards models was performed with the following pre-specified covariates: age, HER2 status (positive vs. negative), molecular subgroup (MMRd vs. *POLE*mut vs. NSMP vs. p53abn), histology (EEC grade 1–2 vs. EEC grade 3 or mixed vs. NEEC), stage (stage I–II vs. III), LVSI (absent vs. present), and adjuvant treatment received (CTRT vs. RT). Median duration of follow-up was estimated by the reversed Kaplan–Meier method [49]. A two-sided *p*-value < 0.05 was considered statistically significant.

## 5. Conclusions

This study is the first to describe a very strong correlation between HER2 positivity and the p53abn molecular subgroup within a cohort of HREC. This finding supports molecular subgroup-directed testing of HER2 status, which is superior to histologic subtype-directed testing. Future clinical trials investigating trastuzumab (or other anti-HER2 therapies) should therefore screen patients with p53abn EC for HER2 positivity to determine eligibility.

## Figures and Tables

**Figure 1 cancers-13-00044-f001:**
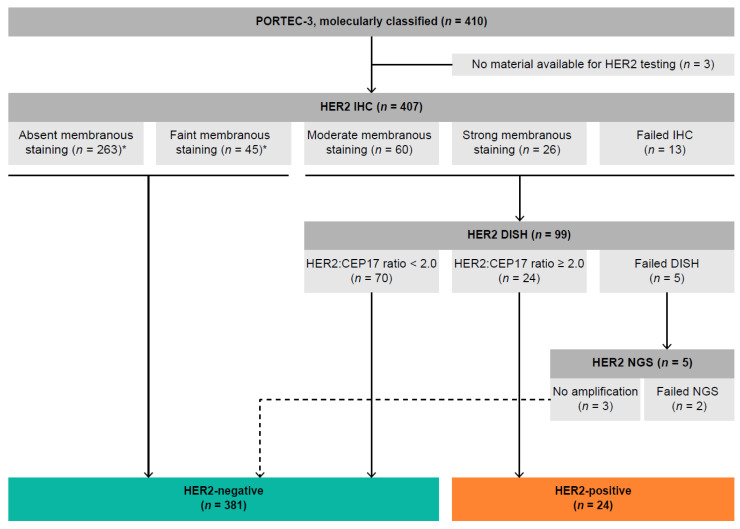
Flowchart of HER2 testing of molecularly classified PORTEC-3 trial participants. * SISH was performed on a random selection of cases with absent (*n* = 10) and faint (*n* = 10) membranous staining to confirm the absence of HER2 amplification. Abbreviations: IHC, immunohistochemistry; SISH, silver in situ hybridization; NGS, next-generation sequencing.

**Figure 2 cancers-13-00044-f002:**
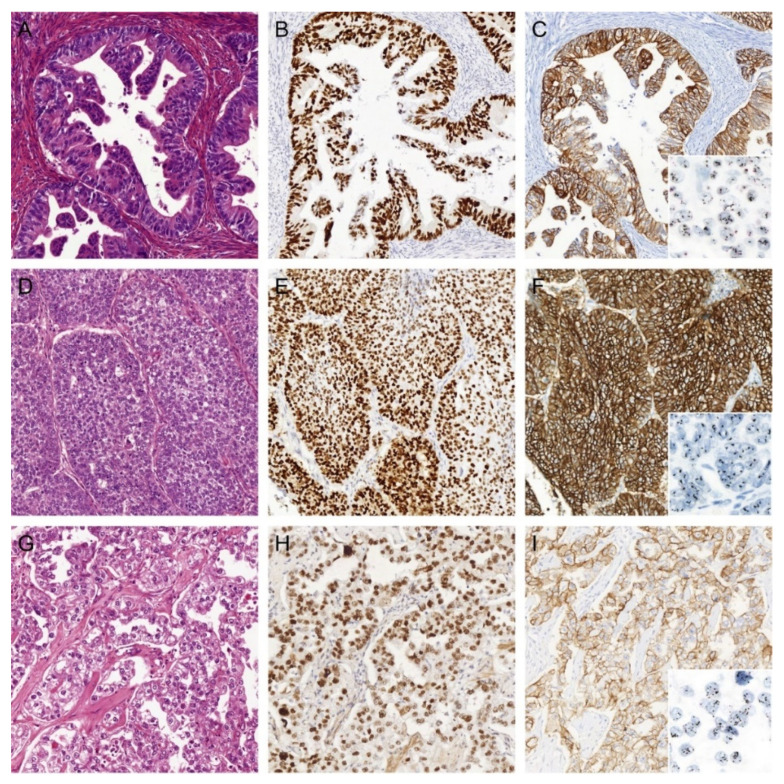
Three examples of HER2-positive endometrial cancers (EC) of non-serous histology. (**A**) H&E of an EC diagnosed as International Federation of Gynecology and Obstetrics (FIGO) grade 2 endometrioid EC (EEC) with (**B**) aberrant mutant-like p53 immunostaining, (**C**) strong (in)complete membranous HER2 immunostaining, and HER2 gene amplification by dual in situ hybridization (DISH). (**D**) H&E of an EC diagnosed as FIGO grade 3 EEC with (**E**) aberrant mutant-like p53 immunostaining, (**F**) strong complete membranous HER2 immunostaining, and HER2 gene amplification by DISH. (**G**) H&E of an EC diagnosed as endometrial clear cell carcinoma with (**H**) aberrant mutant-like p53 immunostaining, (**I**) strong complete membranous HER2 immunostaining, and HER2 gene amplification by DISH. All images ×20 objective magnification.

**Figure 3 cancers-13-00044-f003:**
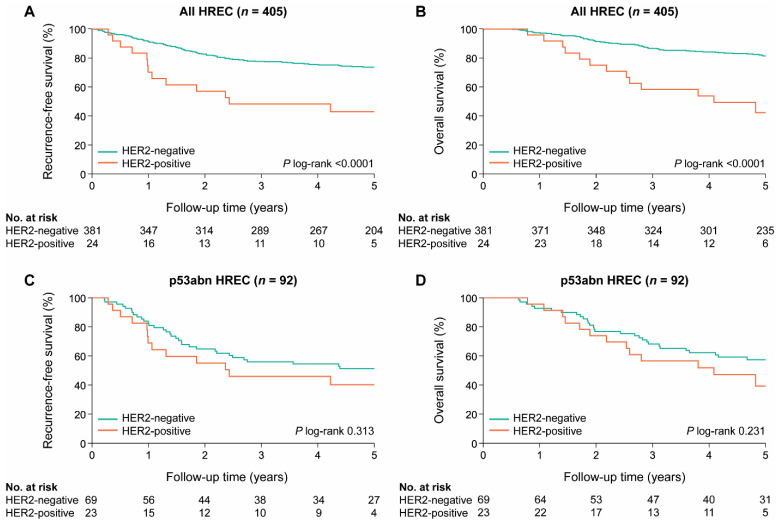
Kaplan–Meier survival curves for 5-year (**A**) recurrence-free survival (RFS) and (**B**) overall survival (OS) for PORTEC-3 patients with HER2-positive and HER2-negative high-risk endometrial cancer (HREC); 5-year (**C**) RFS and (**D**) OS for patients with HER2-positive and HER2-negative p53 abnormal HREC.

**Table 1 cancers-13-00044-t001:** Patient, tumor, and treatment characteristics by HER2 status.

Characteristic	Total	HER2-Negative	HER2-Positive	*p*-Value
	*n* = 405 (100%)	*n* = 381 (94.1%)	*n* = 24 (5.9%)	
Age, years				<0.0001
Mean (range)	61.2 (26.7–80.5)	60.8 (26.7–78.6)	68.3 (55.8–80.5)	
Histotype				<0.0001
Endometrioid	272 (67.2)	266 (69.8)	6 (25.0)	
Serous	64 (15.8)	55 (14.4)	9 (37.5)	
Clear cell	39 (9.6)	34 (8.9)	5 (20.8)	
Mixed (EEC-S)	9 (2.2)	8 (2.1)	1 (4.2)	
Mixed (EEC-CCC)	9 (2.2)	8 (2.1)	1 (4.2)	
Other	12 (3.0)	10 (2.6)	2 (8.3)	
Grade				<0.0001
1–2	163 (40.2)	162 (42.5)	1 (4.2)	
3	242 (59.8)	219 (57.5)	23 (95.8)	
Stage				0.080
IA	53 (13.1)	47 (12.3)	6 (25.0)	
IB	73 (18.0)	68 (17.8)	5 (20.8)	
II	102 (25.2)	96 (25.2)	6 (25.0)	
IIIA	45 (11.1)	44 (11.5)	1 (4.2)	
IIIB	29 (7.2)	27 (7.1)	2 (8.3)	
IIIC	103 (25.4)	99 (26.0)	4 (16.7)	
LVSI				0.38
Absent	152 (37.5)	145 (38.1)	7 (29.2)	
Present	253 (62.5)	236 (61.9)	17 (70.8)	
Lymphadenectomy				0.64
No	184 (45.4)	172 (45.1)	12 (50.0)	
Yes	221 (54.6)	209 (54.9)	12 (50.0)	
Treatment received				0.45
RT	199 (49.1)	189 (49.6)	10 (41.7)	
CTRT	206 (50.9)	192 (50.4)	14 (58.3)	

Abbreviations: EEC-S, mixed endometrioid and serous endometrial cancer; EEC-CCC, mixed endometrioid and clear cell endometrial cancer; RT, radiotherapy; CTRT, chemoradiotherapy.

**Table 2 cancers-13-00044-t002:** Association between the HER2 status and molecular EC classification.

Cohort		Molecular Subgroup				
	Total	*POLE*mut	MMRd	NSMP	P53	*p*-Value
	*n* = 405	*n* = 52 (12.8%)	*n* = 135 (33.3%)	*n* = 126 (31.1%)	*n* = 92 (22.7%)	
PORTEC-3						<0.0001
HER2-negative	381 (94.1)	52 (100.0)	135 (100.0)	125 (99.2)	69 (75.0)	
HER2-positive	24 (5.9)	0 (0.0)	0 (0.0)	1 (0.8)	23 (25.0)	
	Total	*POLE*mut	MSI	CN-low	CN-high	*p*-Value
	*n* = 506	*n* = 49 (9.7%)	*n* = 148 (29.2%)	*n* = 146 (28.9%)	*n* = 163 (32.3%)	
UCEC TCGA PanCancer						<0.0001
Non-*ERBB2*-amplified	481 (95.1)	49 (100.0)	148 (100.0)	146 (100.0)	138 (84.7)	
*ERBB2*-amplified	25 (4.9)	0 (0.0)	0 (0.0)	0 (0.0)	25 (15.3)	

Abbreviations: *POLE*mut, *POLE*-(ultra-)mutated; MMRd, mismatch repair-deficient; NSMP, no specific molecular profile; p53abn, p53-abnormal; TCGA, the Cancer Genome Atlas; UCEC, uterine corpus endometrial carcinoma; MSI, microsatellite-unstable; CN-low, copy number-low; CN-high, copy number-high.

**Table 3 cancers-13-00044-t003:** Multivariable analysis of the HER2 status and clinicopathological features in high-risk endometrial cancers (*n* = 405).

Characteristic		Recurrence-Free Survival			Overall Survival		
			118 Events			92 Events	
	Total n	HR	95% CI	p-Value	HR	95% CI	*p*-Value
Age	405	1.028	1.003–1.054	0.030	1.056	1.025–1.088	<0.0001
HER2 status							
Negative	381	1			1		
Positive	24	1.150	0.596–2.220	0.68	1.237	0.632–2.419	0.54
Molecular subgroups							
MMRd	135	1			1		
p53abn	92	2.720	1.594–4.639	0.000	2.297	1.296–4.071	0.004
*POLE*mut	52	0.085	0.011–0.625	0.015	0.106	0.014–0.789	0.028
NSMP	126	0.984	0.601–1.612	0.95	0.600	0.323–1.112	0.11
Histology and grade							
Endometrioid, low grade	160	1			1		
Endometrioid, high grade	112	1.086	0.637–1.852	0.76	1.353	0.734–2.494	0.33
Non-endometrioid	133	0.842	0.476–1.490	0.55	1.015	0.532–1.936	0.97
Stage							
I–II	228	1			1		
III	177	2.047	1.374–3.048	0.030	1.826	1.174–2.841	0.008
LVSI							
Absent	152	1			1		
Present	253	1.281	0.838–1.957	0.25	1.173	0.720–1.909	0.52

Abbreviations: HR, hazard ratio; *POLE*mut, *POLE*-(ultra-)mutated; MMRd, mismatch repair-deficient; NSMP, no specific molecular profile; p53abn, p53-abnormal; LVSI, lymphovascular space invasion.

## Data Availability

The PORTEC-3 data presented in this study are available on request from the corresponding author. Publicly available The Cancer Genome Atlas datasets were analyzed in this study. This data can be found here: http://www.broadinstitute.org/cancer/cga.

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
