# Peer review of "HER2 Status in High-Risk Endometrial Cancers (PORTEC-3): Relationship with Histotype, Molecular Classification, and Clinical Outcomes"

_cancers, 2020, doi:10.3390/cancers13010044_

Round 1
Reviewer 1 Report
In this study, the authors determine the prevalence of Her2 positivity (defined as Her2 amplification) in a well-characterized cohort of patients with high-risk endometrial carcinoma. This is the first study to correlate Her2 positivity with the molecular subtype, remarkably documenting a strong association between Her2-positivity and p53 abnormal molecular subgroup. An association between Her2 status and survival was observed, which was lost when controlling for molecular grouping or other prognostic variables. The findings are relevant and novel. The manuscript is well-written, organized and concise. I suggest the following points for consideration:
1. I agree with the authors approach not to apply current guidelines for Her2 IHC interpretation in breast or gastric cancer to determine need for DISH testing, as such guidelines have not been thoroughly validated in endometrial cancer. However, I think the authors may be missing an opportunity to provide precisely some degree of validation using their cohort. In other words, it would be useful to classify Her2 IHC following one of the current recommendations (at least the 2007 ASCO/CAP guidelines for breast cancer, which were used in the phase II trial on patients with endometrial serous carcinoma, refs 22 and 23), and determine how many tumours in each IHC category were positive for Her2 amplification. Clinically meaningful thresholds ultimately will require a trial with patients receiving anti Her2-treatments.
2. Related to the previous point, the IHC threshold to trigger DISH testing is rather vague ("moderate" or "strong" membranous staining). My understanding is that the authors wanted to keep a low threshold to capture all the potential cases with amplification. However, the rationale of choosing only intensity of IHC staining, discounting completeness of membranous staining of % of cells, for this purpose is not well explained.
3. Likewise, Figure 1 should include the 20 cases with absent or faint membranous Her2 staining that underwent dual in-situ hybridization. How were these 20 cases selected? Randomly or by their Her2 expression?
4. The two following statements appear contradictory:
Page 4, line 136-37: Incomplete membranous immunoreactivity was seen
in 111 tumours (27.2%), including 11 HER2-positive EC (45.8%).
Page 8, line 227-228: Incomplete membranous HER2 expression was observed in 25% (n=6) of HER2-amplified EC.
5. The findings of this study are compelling in the sense that we can eventually restrict Her2 testing to ECs with abnormal p53 only. To that end, I wonder if the authors were prompted to re-examine the only case in the NSMP group that was positive for Her2. Could this tumour be p53-abn / copy number high? What were the characteristics of this outlier? What was the outcome of the patient?
Author Response
In this study, the authors determine the prevalence of Her2 positivity (defined as Her2 amplification) in a well-characterized cohort of patients with high-risk endometrial carcinoma. This is the first study to correlate Her2 positivity with the molecular subtype, remarkably documenting a strong association between Her2-positivity and p53 abnormal molecular subgroup. An association between Her2 status and survival was observed, which was lost when controlling for molecular grouping or other prognostic variables. The findings are relevant and novel. The manuscript is well-written, organized and concise. I suggest the following points for consideration:
- I agree with the authors approach not to apply current guidelines for Her2 IHC interpretation in breast or gastric cancer to determine need for DISH testing, as such guidelines have not been thoroughly validated in endometrial cancer. However, I think the authors may be missing an opportunity to provide precisely some degree of validation using their cohort. In other words, it would be useful to classify Her2 IHC following one of the current recommendations (at least the 2007 ASCO/CAP guidelines for breast cancer, which were used in the phase II trial on patients with endometrial serous carcinoma, refs 22 and 23), and determine how many tumours in each IHC category were positive for Her2 amplification. Clinically meaningful thresholds ultimately will require a trial with patients receiving anti Her2-treatments.
Thank you for this suggestion. We agree that it may be informative to compare our DISH results with IHC results if we had to apply the validated ASCO/CAP 2007 breast cancer (BC) guideline. We have added this comparison as a new table in the supplement (table S2). It is of interest to note, that if we strictly follow the ASCO/CAP 2007 BC guideline, a substantial number of tumours could not have been classified (e.g. cases with moderate or strong, incomplete membranous staining as well as cases with moderate membrane staining in <10% of tumour cells). In addition, a subset of HER2 amplified EC would be scored IHC 1+, thus considered HER2-negative, due to incomplete membranous staining. These findings reinforce the need of an endometrial cancer specific guideline/algorithm. This point was already discussed in the discussion (lines 264-276).
- Related to the previous point, the IHC threshold to trigger DISH testing is rather vague ("moderate" or "strong" membranous staining). My understanding is that the authors wanted to keep a low threshold to capture all the potential cases with amplification. However, the rationale of choosing only intensity of IHC staining, discounting completeness of membranous staining of % of cells, for this purpose is not well explained.
We agree that the description and rationale of our approach towards subsequent DISH-testing can be explained more thoroughly. To illustrate how we defined the intensity of membranous HER2 staining, we have added a supplementary figure (figure S2) with examples of faint, moderate and strong membranous staining. As rightfully pointed out we had the intention to capture all potential HER2 amplified cases, therefore applying a very low threshold for performing additional DISH-testing. The rational for this approach was based on previous publications looking at HER2 IHC as a surrogate for underlying amplification. HER2-amplified endometrial cancers frequently show incomplete membranous immunohistochemical staining. Therefore, we did not consider complete membranous staining as a criterium for subsequent DISH-testing. Furthermore, we did not want to restrict to an arbitrary threshold for the percentage of stained tumour cells, as this is not yet validated in endometrial cancer. In this predefined approach we did not perform DISH on all tumours with no or only faint membranous staining (with positive on slide control), and classified these as HER2 IHC-negative. The decision for this approach is based on previous experience with diagnostic cases as well as preliminary HER2-DISH results. To confirm this approach we randomly selected 20 of those cases and performed DISH as well. All 20 cases showed no HER2 amplification. In further support of this, during the manuscript preparation, we have performed DISH on the complete set of p53abn EC and no amplifications were found in any of the 44 cases with negative/faint staining. This is now also included in figure 1. As suggested by the reviewer we have further clarified our approach in the method section (paragraph 4.3 and 4.4).
- Likewise, Figure 1 should include the 20 cases with absent or faint membranous Her2 staining that underwent dual in-situ hybridization. How were these 20 cases selected? Randomly or by their Her2 expression?
We randomly selected 10 cases with absent membranous staining and 10 cases with faint membranous staining. To include this point in figure 1, we have included an “Asterix” in the box with absent/faint staining which is explained in the figure legends. In addition, in paragraph 4.4 of the methods section we have added the word ‘randomly’ to the sentence, as well as the number of cases tested per staining intensity category. As mentioned above, we also performed subsequent DISH on all p53abn cases with absent of faint membranous HER2 staining.
- The two following statements appear contradictory:
Page 4, line 136-37: Incomplete membranous immunoreactivity was seen
in 111 tumours (27.2%), including 11 HER2-positive EC (45.8%).
Page 8, line 227-228: Incomplete membranous HER2 expression was observed in 25% (n=6) of HER2-amplified EC.
These statements are indeed contradictory, thank you for noticing. The correct frequency of incomplete membranous HER2 expression is 101 cases (24.9%) in the total cohort and 6 (25%) within HER2-positive cases. We have corrected this error in the manuscript.
- The findings of this study are compelling in the sense that we can eventually restrict Her2 testing to ECs with abnormal p53 only. To that end, I wonder if the authors were prompted to re-examine the only case in the NSMP group that was positive for Her2. Could this tumour be p53-abn / copy number high? What were the characteristics of this outlier? What was the outcome of the patient?
Fully agree that it is important to describe the outlier in more detail, and we have added a description in the results section (paragraph 2.2).
We extensively examined the outlier in detail to assure it was not misclassified. H&E re-review confirmed this was a clear cell endometrial cancer. The p53 IHC was revised and had a convincing wildtype staining pattern. In addition, no TP53 mutation was found by NGS analysis. HER2 IHC in this case showed moderate incomplete membranous staining in >10% of the tumour cells. HER2 DISH was positive for HER2 amplification with a HER2:CEP17 ratio of 2.2 and average HER2 count of 3.7 (scored by two observers, blinded for any of the other results). Finally, no obvious amplification was detected by targeted NGS, however the generated copy number plot was of insufficient quality. The patient was alive and without recurrence after 5 years of follow-up. To summarize, this case is not a misclassified p53abn EC but a NSMP EC according to the WHO 2020 molecular EC classification. The case however showed low-level HER2 amplification with moderate incomplete membranous immunoreactivity in >10% of the tumour.
Reviewer 2 Report
- The title is related with the main research question and objective.
- The study is interesting as it demonstrates the high prevalence of HER-2 mutations in the worst TGCA group. Further studies will clarify the importance of this finding for target therapy.
- In general, I suggest rewriting and restructuring the introduction.
- Simplification of some periods will be required prior to publication (Ex. line 112"now that ; etc)
- In the introduction the explanation of the TGCA molecular classification is inadequate. Can you comment on the histopathological, clinical and prognostic characteristics of each TGCA subgroup?
- Discussion: I suggest to restructure your discussion in order to focus mainly on your main findings.
Author Response
- No comments.
- Thank you.
- Thank you for your suggestion. We have rewritten the paragraph introducing the molecular EC classification. In addition, we have simplified several sentences by breaking them up into two separate sentences.
- We have simplified several periods throughout the manuscript.
- We have extended the description of the molecular EC classification in the introduction, including the histopathological and prognostic characteristics of each group.
- Thank you for the suggestion. We have repositioned the paragraph on the strong correlation with the p53-abnormal molecular subgroup to the beginning of the discussion, as this is our most important finding.
Reviewer 3 Report
The problem described in the article is really relevant. The work is interesting. There are a number of questions for the authors. It is known that overexpression of HER2 is determined in endometrial cancer in 1/3 of cases, is not associated with morphological differentiation of the tumor, depth of myometrial invasion, metastatic lesion of the lymph nodes, however, compared with cases in which there is no overexpression of HER2, it increases the frequency of relapses by 2, 5 times and 1.3 times reduces the results of 5-year survival of patients. However, for this type of cancer, estrogen and progesterone receptors have high prognostic characteristics. I met information that high expression of HER2 in a tumor levels the favorable prognostic value of a positive hormone-receptor status: the 5-year survival rate of patients with endometrioid cancer decreases from 88% to 67%. Do you plan to conduct a similar analysis on your sample of patients? Why was Ki-67 proliferation not evaluated?
Author Response
Indeed positive oestrogen and progesterone receptor-status has shown prognostic value in endometrial cancer, predominantly in low-risk cancers. However, the prognostic value of these receptors in the context of the molecular classification and in high-risk endometrial cancer is less well understood. Due to limited amount of events for 5-year recurrence-free and overall survival, we have chosen to only include established clinicopathological risk features and the molecular classification to investigate the prognostic relevance of HER2-status. However, we do plan to investigate the prognostic value of multiple biomarkers, including the molecular EC classification and ER/PR status, in endometrial cancer in future studies.
In breast cancer, Ki67 proliferation index is a prognostic and predictive biomarker. Here, it is most informative in ER/PR-positive-HER2-negative tumours to predict prognosis and response to chemotherapy. However, in endometrial cancer Ki-67 is not a commonly used prognostic and/or predictive marker. We therefore did not include Ki-67 in our study design.
Round 2
Reviewer 1 Report
Thank you for thoroughly addressing my comments.